# Application of a Multilayer Perceptron Artificial Neural Network for the Prediction and Optimization of the Andrographolide Content in *Andrographis paniculata*

**DOI:** 10.3390/molecules27092765

**Published:** 2022-04-26

**Authors:** Bibhuti Bhusan Champati, Bhuban Mohan Padhiari, Asit Ray, Tarun Halder, Sudipta Jena, Ambika Sahoo, Basudeba Kar, Pradeep Kumar Kamila, Pratap Chandra Panda, Biswajit Ghosh, Sanghamitra Nayak

**Affiliations:** 1Center for Biotechnology, Siksha ‘O’ Anusandhan (Deemed to be University), Kalinga Nagar, Bhubaneswar 751003, Odisha, India; bibhutibhusanchampati77@gmail.com (B.B.C.); pbhuban89@gmail.com (B.M.P.); asitray2007@gmail.com (A.R.); jena.sudipta1991@gmail.com (S.J.); ambikas672@gmail.com (A.S.); basu.cbt@gmail.com (B.K.); pradeepkamila.bapi@gmail.com (P.K.K.); pcpanda2001@yahoo.co.in (P.C.P.); 2Department of Botany, Ramakrishna Mission Vivekananda Centenary College, Kolkata 700118, West Bengal, India; tarunhalder13@gmail.com (T.H.); ghosh_b2000@yahoo.co.in (B.G.)

**Keywords:** *Andrographis paniculata*, andrographolide, ANN model, environmental factors, optimization, prediction

## Abstract

Andrographolide, the principal secondary metabolite of *Andrographis paniculata*, displays a wide spectrum of medicinal activities. The content of andrographolide varies significantly in the species collected from different geographical regions. Therefore, this study aims at investigating the role of different abiotic factors and selecting suitable sites for the cultivation of *A. paniculata* with high andrographolide content using a multilayer perceptron artificial neural network (MLP-ANN) approach. A total of 150 accessions of *A. paniculata* collected from different regions of Odisha and West Bengal in eastern India showed a variation in andrographolide content in the range of 0.28–5.45% on a dry weight basis. The MLP-ANN was trained using climatic factors and soil nutrients as the input layer and the andrographolide content as the output layer. The best topological ANN architecture, consisting of 14 input neurons, 12 hidden neurons, and 1 output neuron, could predict the andrographolide content with 90% accuracy. The developed ANN model showed good predictive performance with a correlation coefficient (R^2^) of 0.9716 and a root-mean-square error (RMSE) of 0.18. The global sensitivity analysis revealed nitrogen followed by phosphorus and potassium as the predominant input variables influencing the andrographolide content. The andrographolide content could be increased from 3.38% to 4.90% by optimizing these sensitive factors. The result showed that the ANN approach is reliable for the prediction of suitable sites for the optimum andrographolide yield in *A. paniculata*.

## 1. Introduction

*Andrographis paniculata* (Burm. f.) Nees, of family Acanthaceae, is an important medicinal herb widely used in as many as 26 Ayurvedic formulations [1]. As the whole plant tastes extremely bitter, it is referred to as the “king of bitters.” It is routinely used in the Ayurveda and Unani systems of medicine [2]. The plant is commonly found in the wild and also cultivated in India, Sri Lanka, South Asia, China, South Africa, and a few other countries [3]. In India, it is grown in Assam, Uttar Pradesh, West Bengal, and southern India [4]. *Andrographis paniculata* possesses a wide spectrum of medicinal properties, such as anticancer, antioxidant, antidiabetic, hepatoprotective, antivenom, antimalarial, anti-HIV, antipyretic, antidiarrheal, antifertility, antimicrobial, and antihyperlipidemic activities [5]. In traditional Chinese medicine (TCM), it is used to cure cold and fever [6] and is enlisted in *National List of Essential Drugs A.D. 1999* in Thailand [7]. The major bioactive constituent of *Andrographis paniculata* is andrographolide, a diterpenoid lactone [8]. Andrographolide is known to inhibit the nuclear activator associated with asthma [9]. It activates the pathway to reduce the risk of thromboembolic disorders [10]. Previously, the anti-inflammatory activity of andrographolide has been established with its molecular mode of action [11]. Andrographolide is known to arrest cells at the G1–S phase, thereby revealing its anticancer activity [12].

The amount of andrographolide in *A. paniculata* varies in different geographical regions. The variation in the content may be due to the climate, soil nutrients, and developmental stages [13,14,15,16,17,18]. A lack of high-yielding genotypes of *Andrographis paniculata* with high andrographolide content is a major bottleneck in the commercial production of this medicinally important herb.

The ANN model is considered as a robust tool in discerning nonlinear complex inter-relationships [19] and is widely used for prediction purpose in various fields of research, yielding better results compared to conventional models [20]. The most commonly used ANN model is multilayer perceptron (MLP) of the feed-forward type. This model works in a manner similar to a human brain, having multiple connections of neurons [21]. The MLP is used in various fields, such as pattern recognition, prediction, classification, and function approximation [22]. There are different layers, i.e., the input layer, the output layer, and at least one hidden layer, with multiple neurons or nodes in each layer [23]. The layers are connected through nodes or neurons with a nonlinear activation function at each node/neuron [24]. MLP takes multiple input data in the form of linear combination and evaluates a single output through a nonlinear transfer function [25]. The ANN model has been applied for the prediction of content of secondary metabolites in relation to the soil and environmental factors in several medicinal plants, such as *Bacopa monnieri*, *Hedychium coronarium*, and *Podophyllum hexandrum* [20,24,26,27].

Therefore, in this study, the ANN approach was used to (i) analyze the effects of various environmental factors and soil nutrients on andrographolide content, (ii) detect the most influencing factors affecting andrographolide content using sensitivity analysis, and (iii) predict the andrographolide content at a new site/location.

## 2. Results and Discussion 

### 2.1. Variation in Andrographolide Content

The quantification of andrographolide in 150 accessions of *A. paniculata* collected from different locations in Odisha and West Bengal was carried out. All samples were processed in an identical pattern and were subjected to HPLC analysis. The HPLC chromatograms were developed for each sample and compared with the standard andrographolide chromatogram. The HPLC run time was set at 45 min, and the andrographolide was detected at a retention factor (R_f_) of 17.8 ± 0.05 min at a wavelength of 223 nm. The representative chromatogram of andrographolide and *A. paniculata* sample is given in Figure 1. The calibration curve of andrographolide was linear, with a correlation coefficient (R^2^) of 0.999 in the range of 10–80 mg/L. It was represented by a regression equation of y = 37.386x + 0.15, where y is the response area and x is the andrographolide concentration. The limit of detection and the limit of quantification of the andrographolide were found to be 1.99 and 6.03 mg/L, respectively. The relative standard deviations (RSDs) obtained for precision and stability analysis were 1.75 and 1.58%, respectively, and were within the admissible range. The average percentage of recovery of andrographolide was found to be 98.91%.

The amounts of andrographolide in different accessions of *A. paniculata* varied from 0.28 to 5.45% on a dry weight basis. The highest content was found in accession AP-9 (5.45%). Nine accessions (AP-5, AP-7, AP-17, AP-26, AP-30, AP-31, AP-41, AP-71, and AP-81) showed a higher content of andrographolide (>4%). The content of andrographolide varied significantly among different locations [13]. In a previous study, the andrographolide content was reported to vary in the range of 1.38 to 3.12% on a dry weight basis among 28 accessions of *A. paniculata* collected from 8 states of India [28]. Similarly, another study, from central India, found not much of morphological variations among *A. paniculata* accessions, but significant variation in the amount of andrographolide was observed depending on the geographical locations of occurrence [15]. The variation in the andrographolide content might be due to the influence of environmental factors and site conditions where the species is grown [29]. The production and accumulation of secondary metabolites is influenced by abiotic factors, such as humidity, temperature, light intensity, nutrients, and water, as they can interfere strongly with plant growth and secondary metabolite production [30]. The variation in secondary metabolites in this study might not be due to the developmental stages as all the samples were collected in a single growing season with similar growth features.

### 2.2. Development of the Artificial Neural Network (ANN) Model

The ANN model was tested by setting the number of neurons in the range of 2–13 in the hidden layer. It is important to set an appropriate number of neurons in the hidden layer to have a best-fit ANN model. Too many neurons will suppress the predictive ability, and too few will hinder the learning process [31]. After a lot of iterations, the ANN model with a topological architecture of 14-12-1 showed the best predictive performance (Figure 2). The 14-12-1 ANN architecture implies 14 independent variables (neurons) in the input layer, 12 neurons in the hidden layer, and 1 dependent variable (neuron) in the output layer. The correlation coefficient (R^2^) between the predicted and observed values in the training, testing, and validation dataset were 0.9741, 0.9631, and 0.9883, respectively (Figure 3). The root-mean-square errors of training, testing, and validation dataset of the developed model were 0.1793, 0.2530, and 0.2190 respectively. An accurate ANN model should have a correlation coefficient (R^2^) nearer to 1 and a root-mean-square error (RMSE) close to 0. The predicted and observed values of training, testing, and validation dataset are given in the Appendix A. Several studies have shown that a single hidden layer can have good predictive performance [24,26,32].

In the last few decades, the ANN model has been used by several researchers for the prediction of drug yield in various medicinal plants. The content of coronarin D, in *H. coronarium*, was predicted through the ANN model [24]. Similarly, the content of podophyllotoxin, the bioactive constituent of *Podophyllum hexandrum*, was assessed through the ANN model [20]. The prediction of the bacoside content in *Bacopa monnieri* was made using the ANN model [26]. The ANN model was also applied for the prediction and optimization of the essential oil yield and the curcumin content in turmeric [27,33].

### 2.3. Sensitivity Analysis

Sensitivity analysis would help to predict the influential environment or soil factor that affects the andrographolide content. The influential parameters were ranked in the decreasing order of their sensitivity (Table 1). The sensitive parameter having the highest error quotient is considered to be the most influential factor. From the study, nitrogen, phosphorus, and potassium were identified as the most influential factors affecting the andrographolide content in *A. paniculata*. In our investigation, nitrogen was found to be the most influential factor, with the highest error quotient, of 5.24, followed by phosphorus and potassium, with error quotients of 3.65 and 2, respectively.

The growth and development of plants depend upon many factors, of which, nutrients play a crucial role. Macronutrients have an important role in determining the growth, yield, and quality of crops [34]. Plants require an optimum level of nutrients for carrying out the cellular processes necessary for their growth and development. This study revealed that the andrographolide content in *A. paniculata* was high in areas where the soil nitrogen content was low. The decrease in the andrographolide content with an increase in soil nitrogen could be due to the production of excess ammonium ions, resulting in the suppression of biosynthesis of secondary metabolites [35]. Andrographolide content and biomass yield have been reported to be maximum when the nitrogen level in the soil was 40 kg/ha [36]. Another study by Radusiene et al. (2019) reported that the nitrogen level decreases the total content of major phenolics in *Hypericum pruinatum* [37]. According to carbon/nutrient balanced hypothesis, when the nitrogen content in the soil is low, the plants limit their growth and reduce the photosynthesis rate, as a result of which, the carbon that is needed for growth is used in the formation of carbon-based secondary metabolites [38]. Phosphorus is the second-most-influencing macronutrient affecting the content of andrographolide. Phosphorus is an essential element in determining the growth and productivity of plants as it is a vital component of nucleic acids, bio-membranes, and energy regulatory system of the cells. In our investigation, it was observed that relatively higher amounts of andrographolide were present in accessions where the soil phosphorus was comparatively high. Phosphorus deficiency in plants reduces their growth, which in turn affects the quality and yield of the crop [39]. Previous studies have found the maximum content of andrographolide when the phosphorus content in soil was 40 kg/ha [40]. Potassium is the third important macronutrient affecting the andrographolide content. Potassium plays a significant role in crop quality, and its deficiency can cause serious problems in plant growth and development. The level of potassium should be maintained high, especially during the developmental stages, to prevent serious loss in crop yield and quality [41]. Potassium balances the electric potential and produces energy-yielding molecules, such as ATP and NADPH, in the chloroplast, which might be responsible for increased andrographolide production [42]. The plant growth continued to increase with an increase in potassium levels in the soil, and the highest growth was observed when the potassium was in the range of 420–560 kg/ha [43]. Our result was in close agreement with the above study, which reported andrographolide content to be the maximum when the level of potassium was in the range of 550–600 kg/ha in the soil.

### 2.4. Optimization of Andrographolide Content

This study revealed that the content of andrographolide can be increased through soil nutrient management by increasing/decreasing the levels sensitive factors, such as nitrogen, phosphorus, and potassium, in the soil. To achieve this, soil nutrients such as nitrogen, phosphorus, and potassium, which have much influence on the andrographolide content, were changed in the ANN model. The nitrogen level was decreased from 59.94 to 34.34 kg/ha and the phosphorus and potassium levels were increased from 35.98 to 41.02 kg/ha and from 409.06 to 600.11 kg/ha in the developed ANN model (Figure 4). After changing the sensitive factors in the ANN model developed, the andrographolide content was 4.90%, which was higher than the andrographolide content (3.38%) obtained from the originally developed ANN model.

The production of secondary metabolites in plants is a complex process that depends upon several factors, such as growth stage; genetic variability; season; climatic, edaphic, and other environmental factors. The variation in the andrographolide content might be due to one or more above related factors, which needs to be investigated in detail.

### 2.5. Application of the ANN Model at a New Site

The developed model was applied at a new site to predict the andrographolide content (Figure 5). *A. paniculata* sample was taken from Athagarh, Cuttack, Odisha, and the soil and climatic parameters were recorded. The soil and climate parameters of the same sample were trained in the developed ANN model for the prediction of the andrographolide content. The experimental value of the andrographolide content obtained from HPLC was 3.48% on a dry weight basis, and the predicted andrographolide content from the ANN model was 3.88% on a dry weight basis. The accuracy of the ANN model in predicting the andrographolide content was found to be close to 90%. Therefore, the developed ANN model could be useful for the prediction of the andrographolide content from samples collected from unknown locations using soil and climatic factors.

## 3. Materials and Methods

### 3.1. Plant Samples and Chemical Reagents

The plant samples were collected in the months of October–November at the maturity stage. The aerial parts of the plants were taken off without uprooting the plants. The geographical coordinates, such as latitude, longitude, and altitude, of each location were noted with the help of a GPS device (Garmin 276 C, Garmin, Olathe, KS, USA). About 500 g of soil sample was collected from each collection site. A total of 150 samples were collected from different geographical regions of Odisha and West Bengal, which are detailed in Appendix A. The collected plants were authenticated by taxonomist Dr. P.C. Panda, Professor, Centre for Biotechnology, Siksha ‘O’ Anusandhan (Deemed to be University), Bhubaneswar, Odisha, India. The required chemicals, such as potassium dihydrogen phosphate, ortho-phosphoric acid, HPLC-grade methanol, and acetonitrile, and water were procured from Merck Life Science Pvt. Ltd., Mumbai, India. Reference-standard andrographolide (Figure 6) of purity >95% was procured from Natural Remedies Pvt. Ltd., Bangalore, India.

### 3.2. Quantification of Andrographolide

#### 3.2.1. Plant Samples and Standard Solution Preparation

The plant samples were washed thoroughly in tap water to remove any dirt. The samples were shade-dried and ground into powder. The powdered samples were passed through a 100 µm sieve to obtain a fine powder. Each sample was accurately weighed to 25 mg, kept in a volumetric flask, and extracted with 50 mL methanol in an ultrasonic bath for 20 min at 60 °C. After extraction, the sample containing the solution was filtered through a 0.22 µm syringe-driven filter for HPLC analysis. Stock solutions of reference-standard andrographolide were prepared in methanol and kept at 4 °C prior to HPLC analysis.

#### 3.2.2. HPLC Instrumentation and Chromatographic Conditions

For HPLC analysis, a Shimadzu HPLC system (Kyoto, Japan) was used, which included a binary LC-20 AD pump, an SPD-20A diode array detector, a CT0-20AC column oven, CBM-20A controller, and a Rheodyne 8125 injector. Chromatographic separation was performed on a Shimadzu Shimpak C18 column (250 mm × 4.6 mm, 5 m) with Solvent A (orthophosphate buffer, pH 2.4) as the mobile phase and a gradient flow from pump A of 95% for 0–17 min, 55% for 18–24 min, 20% for 25–34 min, 55% for 35–39 min, and 95% for 40–45 min. The flow rate remained constant at 1.5 mL/min. The injection volumes for the samples and reference solutions were each set to 20 µL. The run time was set to 45 min. For chromatogram acquisition, the PDA detector was set at 223 nm and UV spectra were obtained between 190 and 800 nm. The mobile phase was passed through a 0.45 µm membrane filter and degassed with an ultrasonic bath before HPLC analysis. The mobile phase and extraction technique for HPLC analysis as optimized by Champati et al. [44] was used in this study.

#### 3.2.3. Quantitative Estimation of Andrographolide

The prepared samples and standard were applied in the HPLC analysis. The retention time and absorbance of the peaks were matched between the standard and plant samples. The identified peaks of andrographolide were integrated, and the areas enclosed by the peaks were noted. The amount of andrographolide was estimated by using a calibration curve (Figure 7) that was generated by plotting the peak area (y) against the concentration (x, mg/l) of the target compound.

The method was validated in terms of recovery, precision, stability, limit of detection (LOD), and limit of quantification (LOQ) as described by Champati et al. [44]. The method of standard addition was used to determine the recovery test, which involved adding a known amount of standard to the analyzed sample and then reanalyzing it. The precision was determined by injecting the standard in replicate solutions three times in one day. The standard’s stability was determined by injecting the same solution for two days at intervals of 24 h (0, 24, and 48 h). The minimum concentration of an analyte at which the peak area of the signal is at least three times greater than the signal-to-noise ratio (S/N ≥ 3) is known as the limit of detection (LOD). The lowest concentration of an analyte at which the peak area is at least 10 times higher than the signal-to-noise ratio (S/N ≥ 10) is known as the limit of quantitation (LOQ).

### 3.3. Quantitative Analysis of Soil and Climatic Factors

The soil was collected from a depth of 0.5–1 m in the vicinity of rhizosphere. About 500 g of soil sample was collected in zip-lock polythene bags from each site. Soil parameters such as organic carbon (OC), nitrogen, phosphorus, potassium, sulfur, pH, and electrical conductivity (EC) were quantified and are listed in Appendix A.

The soil samples were sieved through a 2 mm mesh and then kept for various analyses. The soil samples were suspended in water (1:2::soil:water). The solution was equilibrated by constantly stirring for about 25–30 min, and then the pH was measured. The pH of the soil samples was measured in a Systronics pH meter (Model MKVI). The alkaline KMnO_4_ method was used for estimating the amount of nitrogen in the soil samples [45]. In an 800 mL Kjeldahl flask, 20 g of soil and 100 mL of 0.32% solution of KMnO_4_ was taken. To this, a 2.5% solution of NaOH was added with some distilled water. The mixture was then put into distillation and the distillate was collected in a 250 mL conical flask having 20 mL boric acid with mixed indicator. The distillate was then titrated against 0.02 N H_2_SO_4_, and the amount of available nitrogen was calculated. The available phosphorus was estimated with the help of a calibration curve of standard phosphorus in a spectrophotometer (Thermo scientific evolution 201, Thermo Fisher Scientific, Waltham, MA, USA) at 660 nm. The Bray 2 extractant method was followed for phosphorus quantification [46]. An amount of 5 g of soil was taken in a conical flask along with 25 mL of 1M ammonium acetate. The solution was shaken for 5 min and filtered. The filtrate was tested for potassium estimation with the help of a photoelectric flame photometer (Systronics, model no: 126). The amount of sulfur in soil samples was determined by barium sulfate turbidimetric measurement [47]. A mixture of 0.5 g of soil and 1.4 g of sodium peroxide was prepared and the amount of sulfate was determined with the help of a turbidimeter (Systronics^®^, Type 131). The total organic carbon in the soil samples was determined by the oxidation of organic carbon with potassium dichromate (1.5 N). The reaction took place in an acidic environment of sulfuric acid. The oxidation product was then analyzed in the spectrophotometer [48].

The values of soil factors such as organic carbon, nitrogen, phosphorus, potassium, sulfur, pH, and electrical conductivity vary in the range of 0.11–1.4%, 30–247 kg/ha, 7–49 kg/ha, 29–708.9 kg/ha, 0.26–1670 kg/ha, 4.08–8.37, and 0.01–2.9-Ds/m, respectively.

The climatic factors of the sample collection sites were collected from weather stations located at or close to those sites. The factors included average temperature (24–27.6 °C), minimum temperature (9.7–16.6°C), maximum temperature (32.2–42.1°C), annual rainfall (704–3274 mm), annual relative humidity (49.42–80.36 %), and ultra-violate radiation index (6–7.34), which were taken from https://en.climate-data.org/ and https://www.worldweatheronline.com/ (accessed on 15 February 2020) for the year 2018–2019. The data for climatic factors are given in the Appendix A.

### 3.4. Artificial Neural Network Development

This investigation used the ability of the artificial neural network (ANN) to predict the amount of andrographolide by critically analyzing the soil and climatic factors. The ANN model was executed using STATISTICA 11.0 software (Tibco Stat Inc., Version 13.5.07) with the feed-forward back-propagation (BP) method. The method was trained using soil and climatic factors as input and andrographolide as output. The optimal topological structure of the ANN model was determined by a trial-and-error method. The best-fit architectural model was selected from 10,000 possible structures. The ANN model’s performance was assessed by measuring parameters such as coefficient of determination (R^2^), root-mean-square error (RMSE), mean absolute error (MAE), and mean absolute percentage error (MAPE). To establish the best-fit model with the appropriate hidden layer and neuron, different numbers of hidden layers (1–5) and the number of neurons in a hidden layer (2–13) were tested. The independent variables are altitude, relative humidity, UV index, average temperature, minimum temperature, maximum temperature, annual precipitation, pH, organic carbon, electrical conductivity, nitrogen, phosphorus, potassium, and sulfur. Andrographolide was the dependent parameter. The whole structure of the ANN model consists of an input layer (independent variables), a hidden layer, and an output layer (dependent variable). A total of 150 accessions were taken for ANN modeling. The samples were divided into three groups for training, testing, and validation purposes in the ANN dataset. The samples were randomly distributed into 70% (106 samples) for training, 15% (22 samples) for testing, and 15% (22 samples) for validation of the ANN model. The hit-and-trial approach was followed for developing the best ANN model by selecting the neurons in the range of 2–13. The number of neurons were optimized, and the perfect neuron combination was selected to achieve the best possible predictive performance. The training process of the dataset regulates different weights and biases to reduce the output value difference between experimental data and predicted data. The neurons at the hidden and input layers manage and adjust the dataset to give the optimum output. These functions were carried out by using the following formulas:$$Y_{s}=Y_{s}=\sum_{t=1}^{p}V_{s}U_{st}+X_{t}$$

Here:*Y_s_* = input to node t in the hidden or output layer;
*s* = number of neurons;
*V_s_* = output of the preceding layer;
*U_st_* = weight interconnection in between the *s* and *t* node;
*X_t_* = bias connected with node *t*.

To measure the nonlinear relationship between the datasets, the sigmoid transfer function was used in the ANN model development. The formula is given below.
$$W_{t}=\frac{1}{1+e^{-ys}}$$
where *W_t_* = output of node *t*.

The function of sigmoid transfer falls between 0 and 1. All the variables were expressed in 0–1 values with the help of the following formulas. This process is called normalization of datasets.
$$X_{norm}=\frac{X-X_{min}}{X_{max}-X_{min}}$$

Here:*X_norm_* = normalized value of the variable; 
*X* = variable;
*X_max_* = highest valued variable;
*X_min_* = lowest valued variable.

The other factors that were taken into consideration during the development of the predictive performance of the model are root-mean-square error (*RMSE*), mean absolute error (*MAE*), and mean absolute percentage error (*MAPE*). They are computed by the following formulas:$$RMSE=\sum_{i=1}^{n}{\left(X_{i}-X_{ik}\right)}^{2}$$
$$MAE=\frac{1}{n\;}\sum_{i=1}^{n}\left|X_{i}-X_{ik}\right|$$
$$PMAE=\frac{1}{n\;}\sum_{i=1}^{n}\left|X_{i}-X_{ik}\right|\times 100$$
where *x_i_* = predicted value, *x_ik_* = experimental value, *x_z_* = mean of experimental value, and *n* = number of observations.

## 4. Conclusions

Geographical location, climatic, and edaphic factors, such as soil nutrients, affect the composition of secondary metabolites in plants. Andrographolide, being the major bioactive phytoconstituent of *A. paniculata,* is influenced largely by the various environmental and edaphic factors. Therefore, it is important to investigate the role of environmental and edaphic factors on the andrographolide content. The multilayer perceptron artificial neural network (MLP-ANN) model was used to analyze the influence of these abiotic (environmental and soil) factors on andrographolide yield and to predict suitable sites for the cultivation of *A. paniculata* with a high andrographolide content. The developed ANN model with the 14-12-1 topology showed the highest correlation coefficient, of 0.972, between predicted and observed datasets. By optimizing the changeable sensitive factors (nitrogen, phosphorus, and potassium content) in the developed ANN model, the andrographolide content was enhanced from 3.38 to 4.90% in *A. paniculata*. Therefore, designing effective soil nutrient management strategies is of great importance for producing high-quality *Andrographis paniculata* for industrial applications.

## Figures and Tables

**Figure 1 molecules-27-02765-f001:**
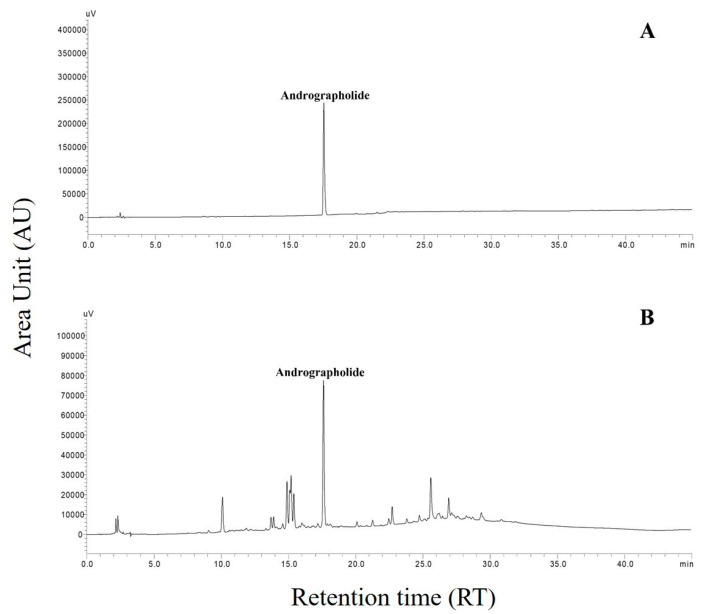
HPLC chromatogram of andrographolide (**A**) and *Andrographis paniculata* (**B**).

**Figure 2 molecules-27-02765-f002:**
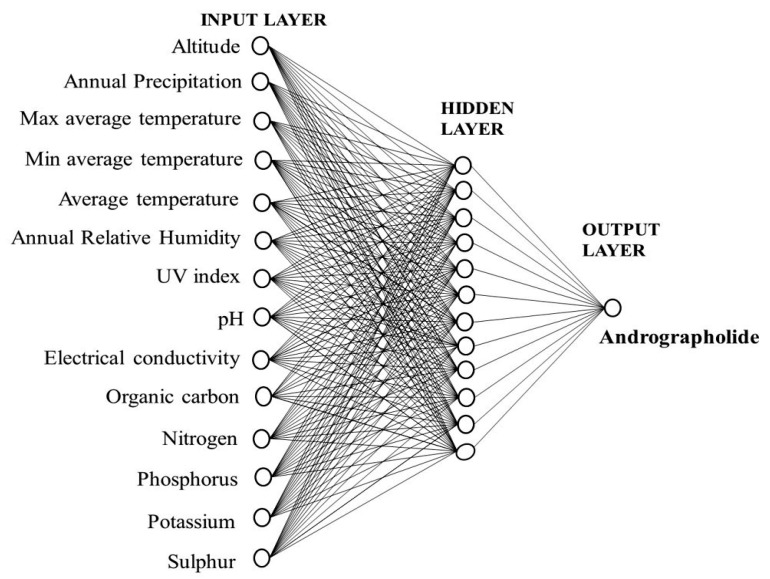
Architecture of the multilayer perceptron feed-forward network used in the study.

**Figure 3 molecules-27-02765-f003:**
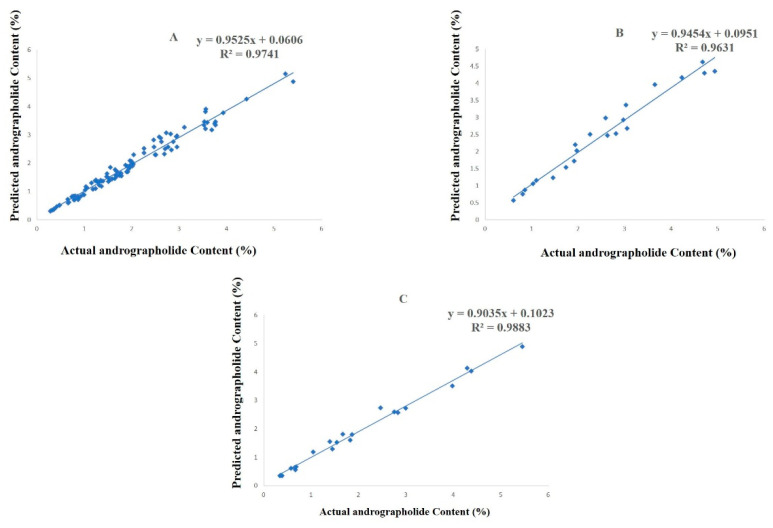
Scatter showing experimental and predicted values of andrographolide in training (**A**), testing (**B**), and validation (**C**).

**Figure 4 molecules-27-02765-f004:**
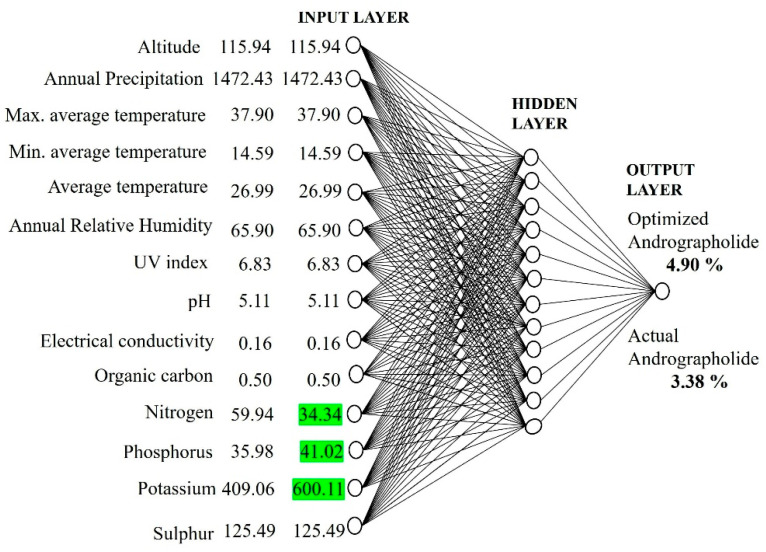
Optimization of andrographolide content by changing input parameters of the ANN model.

**Figure 5 molecules-27-02765-f005:**
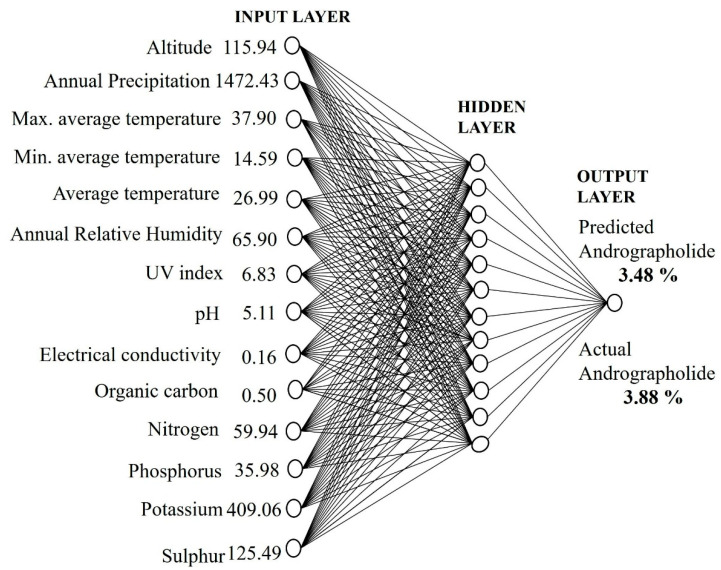
Application of the ANN model at an unknown site for prediction of andrographolide content.

**Figure 6 molecules-27-02765-f006:**
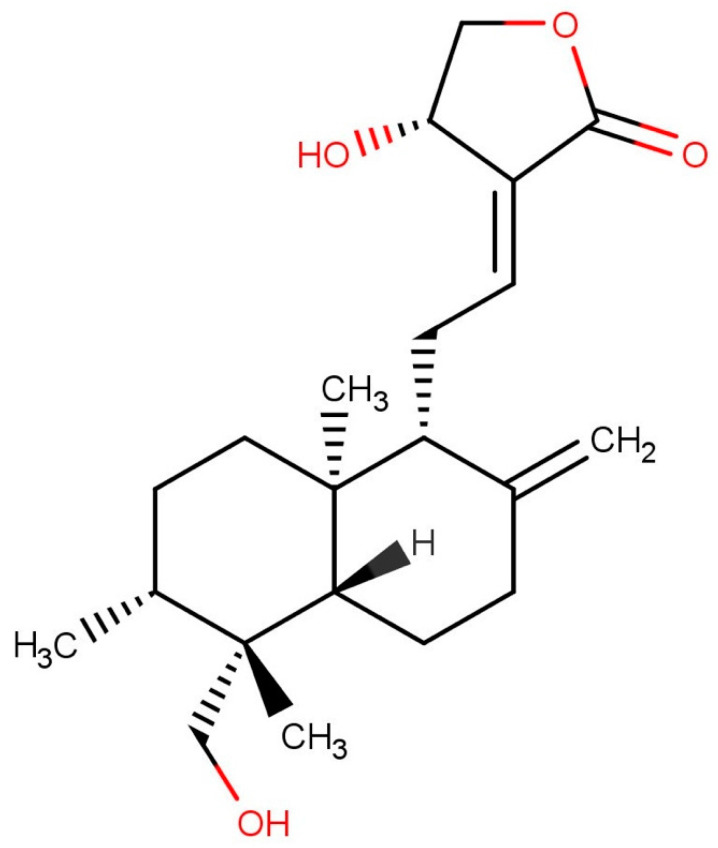
Chemical structure of andrographolide.

**Figure 7 molecules-27-02765-f007:**
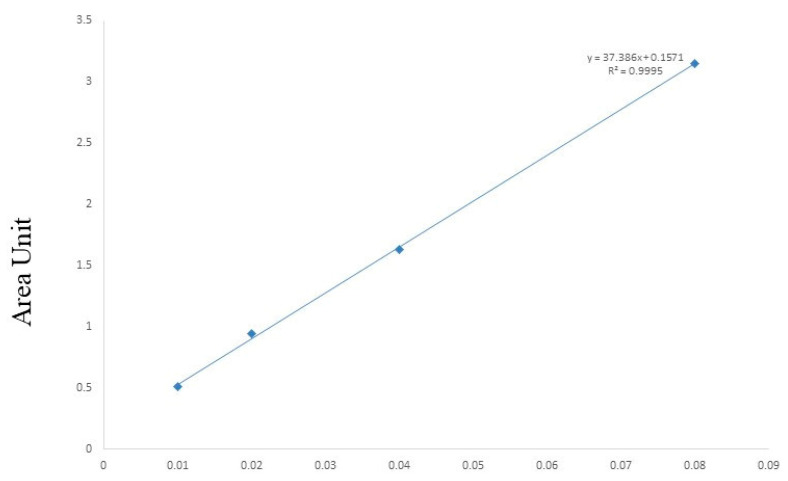
Calibration curve of andrographolide for quantitative analysis.

**Table 1 molecules-27-02765-t001:** Sensitivity analysis of the neural network.

Parameters	Error Quotient	Rank
Nitrogen (kg/ha)	5.247	1
Phosphorus (kg/ha)	3.660	2
Potassium (kg/ha)	2.009	3
Sulfur (kg/ha)	1.038	4
pH	1.038	5
Electrical Conductivity (Ds/m)	1.029	6
Annual Precipitation (mm)	1.018	7
Organic Carbon (%)	1.009	8
UV Index	1.006	9
Annual Relative Humidity	1.004	10
Average Temperature (°C)	1.003	11
Maximum Temperature (°C)	1.002	12
Altitude (m)	0.999	13
Minimum Temperature (°C)	0.993	14

## Data Availability

The data presented in this work are available in the article.

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
