# Peer review of "Application of a Multilayer Perceptron Artificial Neural Network for the Prediction and Optimization of the Andrographolide Content in Andrographis paniculata"

_molecules, 2022, doi:10.3390/molecules27092765_

Round 1
Reviewer 1 Report
Figure quality could be improved.
Long tables (1, 2, 3, 5, 6, 7) could be moved to a supplementary file. For each field / column, a range of values could be added to the main text.
In 2.4. Optimization of andrographolide content, how were the initial and the final values for nitrogen, phosphorus and potassium selected?
Author Response
Response to reviewer 1 comments
Comments to the Author
Figure quality could be improved.
Response: The figure quality has been enhanced to 1200 dpi in the revised manuscript.
Comments to the Author
Long tables (1, 2, 3, 5, 6, 7) could be moved to a supplementary file. For each field / column, a range of values could be added to the main text.
Response: The tables 1,2,3,5,6 and 7 has been moved to supplementary file. The range of values were added in the manuscript wherever applicable.
Comments to the Author
In 2.4. Optimization of andrographolide content, how were the initial and the final values for nitrogen, phosphorus and potassium selected?
Response: The description has been reframed to clearly state how the initial and final values of nitrogen, phosphorus and potassium were selected for optimization study.
Reviewer 2 Report
Authors propose a multilayer perceptron artificial neural network to determine the factors affecting the andrographolide content in Andrographis paniculata.
The manuscript is well organized and the results are critically discussed.
I suggest to revise the submission, taking particular care to:
- keywords the number of keywords should be increased to help serching
- extraction procedure even if reference 44 cites Champati et al., more details about the quantification procedure of % extracted from the plant (recovery) should be detailed
- Tables are too long I suggest to add all the tables as supplementary material
- authors generally report 2018-2019 monthly average of climatic factors. I suggest to state the year each sample was taken, simply by adding a column to one of the tables
- check english (minor revision)
Author Response
Response to reviewer 2 comments
Authors propose a multilayer perceptron artificial neural network to determine the factors affecting the andrographolide content in Andrographis paniculata.
The manuscript is well organized and the results are critically discussed.
I suggest to revise the submission, taking particular care to:
Comments to the Author
Keywords- the number of keywords should be increased to help searching
Response: 3 more keyword has been added
Comments to the Author
Extraction procedure-even if reference 44 cites Champati et al., more details about the quantification procedure of % extracted from the plant (recovery) should be detailed
Response: The method has been elaborated and added in the appropriate section in the manuscript
Comments to the Author
Tables are too long I suggest to add all the tables as supplementary material
authors generally report 2018-2019 monthly average of climatic factors. I suggest to state the year each sample was taken, simply by adding a column to one of the tables
Response: Thank you for your suggestion. The year of samples collection has been added to Table 5 of supplementary file.
Comments to the Author
Check English (minor revision)
Response: The grammar and syntax error has been checked thoroughly.
Round 2
Reviewer 2 Report
The manuscript has been sufficiently improved. I suggest publication in Molecules